# Autologous Tooth Dentin Graft: A Retrospective Study in Humans

**DOI:** 10.3390/medicina58010056

**Published:** 2021-12-30

**Authors:** José Manuel Cervera-Maillo, David Morales-Schwarz, Hilde Morales-Melendez, Lanka Mahesh, José Luis Calvo-Guirado

**Affiliations:** 1Private Practice, 34001 Palencia, Spain; 2Private Practice, 47006 Valladolid, Spain; dgms65@gmail.com (D.M.-S.); hildemoralea90@gmail.com (H.M.-M.); 3Private Practice, New Delhi 110002, India; drlanka.mahesh@gmail.com; 4Facultad Ciencias de la Salud, Universidad Autónoma de Chile, Temuco 8320000, Chile; jose.calvo@uautonoma.cl; 5Private Practice Murcia, 30002 Murcia, Spain

**Keywords:** autogenous particulate dentin graft, autologous graft, bone grafts, human teeth, smart dentin grinder, dental graft

## Abstract

*Background and Objectives*: The aim of this study is to evaluate the efficacy of an autologous dentin graft, via extracted teeth that are processed into bacteria-free particulate dentin in a Smart dentin grinder and then grafted immediately into alveolus post extraction or into bone deficiencies. *Materials and Methods:* Ten healthy, partially edentulous patients with some teeth in the mandible were recruited in the study. After their own teeth were grinded, particulate teeth were placed in empty sockets and bone defects after teeth extractions. Furthermore, after three, six, 12 and 24 months, core samples using a 3 mm trephine were obtained. *Results:* At three months, the particles of grinded tooth were immersed inside a new connective tissue with a small new bone formation (16.3 ± 1.98). At six months, we observed small particles of dentin integrated in new immature bone, without inflammation in the soft tissue (41.1 ± 0.76). At twelve months, we observed a high amount of bone formation surrounding tooth particles (54.5 ± 0.24), and at twenty-four months, new bone, a big structure of bone, was observed with dentin particles (59.4 ± 1.23), statistically different when compared it with at three months. *Conclusions:* A particulate dentin graft should be considered as an alternative material for sockets’ preservation, split technique, and also for sinus lifting. One of the special characteristics after 24 months of evaluation was the high resorption rate and bone replacement without inflammation. This material could be considered as an acceptable biomaterial for different bone defects due to its osteoinductive and osteoconductive properties

## 1. Introduction

The human tooth is composed of 80% dentine, 15% enamel, 5% cement and pulp. Dentin, which is the major component, has a composition very similar to bone thanks to its high mineral phase and its tubular architecture [1]. The morphogenetic protein of dentin can induce new bone formation and be replaced by new bone without causing an inflammatory reaction [2]. Biomaterials are used to repair hard and soft tissue defects; synthetic materials are relatively inexpensive and do not involve a biological risk, but they do not have the capacity for osteoinduction and osteogenesis, so their use is limited to the formation of useful bone. Dental extraction is one of the most performed clinical procedures, but extracted teeth are still considered wasteful and biologically useless, discarding their use [3].

Therefore, thanks to its tubular structure the particulate tooth, as graft material, is highly osteoinductive, Bone Morphogenic proteins (BMPs) give it a high osteoconductive capacity, and type I collagen present in the microtubules favors healing [4]. t is an excellent alternative to autologous bone and seems to have a better tissue response than heterologous and synthetic grafts [4,5,6,7,8,9]. Dentin serves as an autograft because of its composition being almost identical to that of human bone in terms of calcium ions and organized phosphorus such as hydroxyapatite and TCP. Its organic phase contains abundant type I collagen and growth factors. Its organization in microtubules that allow bone growth and favor osteoconduction makes dentin be considered as material for grafts, with a behavior superior to that of xeno-derivatives or other types of grafts [8]. Dentin has inducing properties, and it fuses and gradually replaces and produces bone neoformation [9]. A study in 2013 shows that in the histological analyses signs of a new bone formation appeared at two weeks and that this was because dentin induced the formation of new bone by NCP (non-collagen proteins) embedded in the dentin [10]. In a study from 2018, it was confirmed that the chemical composition of the particles of the tooth crushed with the Smart dentin grinder was clearly like natural bone [11]. Another study in dogs confirms the formation of immature bone and lamellar bone in relation to the healing of the unfilled alveolus and tells us that substantially more bone formation was found in the areas where dentin was placed, generating large amounts of new bone tissue formation after 60 days and small amounts of lamellar bone after 90 days of healing [12]. Thus, several authors have shown that the properties of crushed tooth could act as a bone substitute induced by dentine and dentin pulp, via studies of the recycling of human teeth as a new graft material for bone regeneration [8,13,14]. 

Therefore, based on the scientific evidence that exists on a particulate autologous tooth, we performed this histological study in humans to check for bone neoformation at three, six, 12 and 24 months and see the amount of new bone that was formed and residual graft and connective tissue that was remnant.

## 2. Materials and Methods:

This prospective clinical trial was approved by the Bioethical Committee of the Catholic University of Murcia, Spain (Number 7527/2019. Code CE111901).

### 2.1. Patients

Ten healthy patients in need of a full-arch treatment in the mandible were consecutively recruited. Six women and four men, with a mean age of 64 years (range 44–86), were selected for the treatment of their atrophic edentulous mandible. Each person was informed of the general requirements and purposes of the study, as well as the nature of the planned treatment and the alternative procedures. Based on a multisided CBCT scan, the patient was included if six implants could be placed in the mandible without the need for bone augmentation.

The potential risks, possible complications, and benefits of the proposed treatment were explained to the study patients. All the information was provided in written and oral form. In addition, all the patients signed an informed consent.

In this randomized controlled trial, 30 patients were subjected with indications of tooth extractions and implant placement.

**Inclusion criteria**:Aged ≥ 25 years and committed to participate in up to a 3-month follow-up.Upper and lower teeth with indications to be extracted.


**Exclusion criteria:**



**Systemic:**
Human immunodeficiency virus infection.Presence of metabolic, endocrine, blood, neoplastic, or renal diseases.Alcoholism or drug abuse.Smoking >10 cigarettes per day.Any other conditions that might interfere with the analysis of the results.



**Local:**
Previous bone grafting.History of irradiation therapy.Gum diseases.Bruxism/clenching.Inadequate oral hygiene.Lack of primary stability.Insufficient bone or any abnormality that would contraindicate implant placement.


The patients were divided in 2 groups: (a) Post-extraction sockets and implant gap grafted with dentin graft material, and (b) natural healing sockets, in all cases used in the immediate post-extraction implant protocol. 

The radiographs and CBCT scanner were performed in all patients before and after extraction. The sockets were randomly assigned by the www.radomization.com (accessed on 10 January 2019) program depending on the clinical case of each patient. The patient selection was randomized due to it being a private practice, and patients that came for an extraction were selected to become a study patient after an informed consent was signed. The surgical procedure was anesthesia, extraction, and suture.

Then, the teeth were cleaned with a surgical bur, taking away the periodontal ligament and enamel. The clean and dry tooth, mostly dentin, is immediately grinded using a specially designed Smart Dentin Grinder (Kometabio, Fort Lee, NJ, USA). The teeth were placed inside the Dentin Grinder machine one by one, depending on the clinical case, one tooth at a time. After this, the crushed tooth was placed as a filling material inside the alveolus. The pulps were not grinded because the machine could not grind soft tissue, and the pulps had to be discarded ([Fig medicina-58-00056-ch001]).

The size of the particles was between 300 and 1200 μm in diameter. The sorted particulate dentin was immersed in basic alcohol cleanser in a sterile container for 7 min to dissolve all organic debris and bacteria. Then, the particulate was washed with sterile saline for 3 min. The bacteria-free particulate dentin was ready for immediate grafting into extraction sites or into bone defect sites.

We removed the excess saline solution and mixed our autologous dentin graft with platelet-rich plasma. The bacteria-free particulate dentin was ready to be placed as an immediate graft in the post-extraction socket or at the sites of bone defects.

The biopsy samples were obtained at random at 3, 6, 12 or 24 months. They were taken in the immediate implants with a gap that were filled with dentin and in the area of the post-extraction socket ([Fig medicina-58-00056-ch002]). The extraction of the sample was done using a 3.0 mm trephine bur (Figure 1 and Figure 2).

### 2.2. Histological Preparation

The bone cores containing a mixture of 5% glutaraldehyde and 4% formaldehyde were fixed for three days.

Biopsies were processed for the ground sectioning according to the histological methods. Briefly, samples were dehydrated in increasing grades of ethanol up to 100%, infiltrated with methacrylate, polymerized, and sectioned at the buccolingual plane using a diamond saw (Exakt Apparatebeau, Norderstedt, Hamburg, Germany). Each block was sectioned with a high-precision diamond disk at about a 100 mm thickness and ground to approximately a 50 mm final thickness with an Exakt 400 s CS grinding device (Exakt Apparatebau, Norderstedt, Hamburg, Germany). Sections were stained with hematoxilin-eosin, and a semi-quantitative evaluation of new bone formation, connective tissue and remaining grafts was performed. To obtain a single digitally processable overview image of the cores per site, four images of the same area were taken, selecting the best one.

A histological and histomorphometric study was performed to evaluate the formation of new bone and the behavior of the dental graft and time of resorption.

At three months, the particles of grinded tooth were immersed inside a huge and new connective tissue with a small amount of new bone formation. The particles were still inside without inflammation in the soft tissue (Figure 3 and Figure 4).

At six months, soft tissue integration and a small amount of tooth graft are included (Figure 5 and Figure 6). We can see small particles of dentin integrated in the bone. There is a large amount of transforming soft tissue in the new immature bone, without inflammation in the soft tissue.

At twelve months, we observed a large amount of bone formation surrounding tooth particles. We can see small particles of dentin graft with connective tissue transforming into immature bone without inflammation (Figure 7 and Figure 8).

At 24 months, new bone, a big structure of bone, was observed with the dentin particles, with a small amount of soft tissue and fewer dentin particles inside. Immature bone was observed in some areas, with lamellar bone around the particles. These images were observed due to the dentin particles’ resorption (Figure 9 and Figure 10).

At 24 months, we can see the formation of lamellar bone, how dentin is integrated in the bone and how we have a formation of new bone and mature bone. We use the staining (name of the staining) and the image has a magnification of (magnification of the image) (Figure 11).

## 3. Results

### 3.1. Histological Analysis Showed Signs of New Bone Formation at 60 Days

In the histomorphometric analysis, a new bone formation was observed to increase according to the passage of time, and the residual graft was reduced together with the remaining connective tissue, which remained stable for the year (Table 1). At the same time, a great bone resorption of the dentin graft was observed on the new bone formation.

### 3.2. Histologic Data Obtained after Samples Analysis at 24 Months of Evaluation

In the experimental group, we see a bone formation around implants and the extractions cavities when compared with the control group. Implant placement was possible as early as three months after placing the autogenous dentin graft. There were no complications in the healing.

## 4. Discussion

The current results suggest that autogenous mineralized dentine particles can be considered as an alternative graft material for the preservation of cavity bundle bone, bone augmentation in the breast, or for the repair of bone defects [11]. This 2014 study showed the obtained results, and the authors saw that, as an autogenous material, dentin had physicochemical structures and characteristics that were more similar to autogenous cortical bones [15]. As a material, dentin can be a substitute to autogenous bone for alveolar treatment and bone defects [5]. Dentine particles can be considered a potential material for bone regeneration due to their chemical composition and the amount obtained, since after grinding the teeth, the resulting material increases in quantity up to three times its original volume, so that two incisors teeth, the extracted mandibular laterals, provide a sufficient amount of material to fill four empty mandibular sockets [11]. It also serves as an alternative for patients who do not want allografts and xenografts, providing an excellent biocompatibility without causing an immune response, contagion, or reaction to a foreign material [16,17]. The autogenous mineralized dentin particles grafted immediately after the extractions can be manufactured in various sizes and have osteoinduction, osteoconduction and progressive replacement [18,19,20]. The Bio-Oss particles were not resorbed but became surrounded by new bone; Placement of the biomaterial in the fresh extraction socket retarded healing. Regarding this, we can explain that the grafted extraction sites may not undergo a dimensional change [21]. There are studies that show us that variations in the physical properties of a substitute bone material clearly influence the degradation process, and consequently, biomaterials can be designed on demand depending on the needs of resorption, dimensional stability and management that are necessary for each case [13]. There are also surgeries in which a part of the root is deliberately left to preserve the alveolar bone [22,23]. The clinical relevance of this will be that the use of a dentin graft could be successful, with an immediate placement of temporary or temporary TR implants meeting the requirements for placing a cemented fixed immediate prosthesis, avoiding removable provisional prostheses [21,24]. The particulate dentin graft could help clinicians protect soft bone, augment insufficient areas, and could also lead to a best option for healing a wound quickly and better than alloplastic materials.

## 5. Conclusions

Particulate dentin grafts should be considered as an alternative material for sockets’ preservation, split technique and sinus lifting. One of its special characteristics after 24 month of evaluation was the high resorption rate and bone replacement without inflammation. Dentin particles had open tubes that allowed capillaries to access the inside of them and that allowed a quick resorption. Clinically and histologically, the performance of the dentin graft is at least comparable to extensively used xenogeneic or allogenic biomaterials. Further studies are necessary to confirm these data.

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
