# Peer review of "Autologous Tooth Dentin Graft: A Retrospective Study in Humans"

_medicina, 2021, doi:10.3390/medicina58010056_

Round 1

Reviewer 1 Report

This study aimed to investigate the effects of particulate dentin used for bone graft in clinic. The socket after tooth extraction was filled with the particulate dentin and the histological image of the filled socket was observed. Results showed that bone tissue gradually increased, while the particulate dentin and connective tissue decreased over the period of 24 months. However, authors did not show a reliable data and ethical approval. The design of the experiment is unclear what is the inclusion and exclusion criteria for patients, and the conduct of ethical review by committee was missing. The surgical procedure and how the authors choose extraction site were doubtful. Furthermore, the histological process was not explained how the sample was prepared and how they were evaluated, especially 30 samples from the patients. Result also lacked of reliable data. In the table, it should show a clear parameter that author want to describe and show standard deviation if the number is from 30 slides analysis. Likewise, the histological image did not show scale bar and highlight the area of new bone formation. Author only showed the high magnification image without lower magnification to locate the area of tissue displayed in the histological image. Therefore, conclusion of this study is weak to say that the particulate dentin is useful in clinic and even comparable to other bone substitute materials. Further improvement of this study can be done by including the control group (extraction site without graft materials or with the others bone substitute material).

Author Response

Dear Reviewer

Thanks a lot for your comments. Followed your recommendations and the paper will improve a lot

All the best

Dr. Jose Manuel cervera Maillo

Reviewer 2 Report

The aim of this study is to evaluate the efficacy of autologous dentin graft, extracted teeth that are processed into a bacteria-free particulate dentin in a smart dentin grinder and then grafted immediately into alveolus post extraction or in bone deficiencies.

Comments:

The abstract of the study must contain other information about the methodology used, the statistical data and the results obtained. The introduction should contextualize the current data in contemporary literature, making clear the evidence of a strong justification for the development of the study. The methodology is fragile, and must be described with many additional details (i.e., age, sex, origin of patients, study approval by the ethics committee, type of radiograph and method of acquisition of CBCT, details of microscopic analysis). The legends of histological sections must be improved. The discussion can be much better and bring information related to the study results, methodology, correlations and clinical implications and clinical relevance. The conclusions should only answer the target problem. The references must be standardized according to the Journal.

Author Response

(The authors gave the same response as above.)

Reviewer 3 Report

The authors are requested to add a diagram describing exactly how the sample was obtained and processes till examining the histological section.

How did you separate the cementum and enamel from the obtained samples?

What was the reason for extraction of the teeth?

Was the pulpal tissues included during crushing of the teeth?

Describe accurately how did you sterilize the powdered specimens from possible bacterial infection prior to its implantation in the host site.

Author Response

(The authors gave the same response as above.)

Round 2

Reviewer 1 Report

The manuscript is improved in many aspects. However, there are still some errors in the text and concern on the experimental design. Many area need English editing. Additionally, in order to conclude as the authors suggested, the control group of the experiment might be necessary.

Author Response

Dear Reviewer 

Thanks a lot of your comments

All the best

Reviewer 3 Report

The authors are requested to provide diagramatic illustration for the steps carried out to obtain the dentin as it is not clear how was the pulp tissue separated from the dentin.

The caries teeth utilized as mentioned by the authors cannot be sterilized by merly adding simple disinfectants for short periods.

Author Response

Dear Reviewer 

Thanks a lot for your recommendations

Best regards

Dr. Jose Manuel  Cervera Maillo
